# Machine-Learning-Based Radiomics MRI Model for Survival Prediction of Recurrent Glioblastomas Treated with Bevacizumab

**DOI:** 10.3390/diagnostics11071263

**Published:** 2021-07-14

**Authors:** Samy Ammari, Raoul Sallé de Chou, Tarek Assi, Mehdi Touat, Emilie Chouzenoux, Arnaud Quillent, Elaine Limkin, Laurent Dercle, Joya Hadchiti, Mickael Elhaik, Salma Moalla, Mohamed Khettab, Corinne Balleyguier, Nathalie Lassau, Sarah Dumont, Cristina Smolenschi

**Affiliations:** 1Biomaps, UMR1281 INSERM, CEA, CNRS, Université Paris-Saclay, 94805 Villejuif, France; Samy.AMMARI@gustaveroussy.fr (S.A.); Mickael.ELHAIK@gustaveroussy.fr (M.E.); Salma.moalla@gustaveroussy.fr (S.M.); Corinne.BALLEYGUIER@gustaveroussy.fr (C.B.); Nathalie.LASSAU@gustaveroussy.fr (N.L.); 2Department of Imaging, Gustave Roussy, Université Paris Saclay, 94805 Villejuif, France; Joya.HADCHITI@gustaveroussy.fr; 3Department of Medical Oncology, Gustave Roussy Cancer Campus, 94805 Villejuif, France; TAREK.ASSI@gustaveroussy.fr (T.A.); MOHAMED.KHETTAB@gustaveroussy.fr (M.K.); SARAH.DUMONT@gustaveroussy.fr (S.D.); Cristina.SMOLENSCHI@gustaveroussy.fr (C.S.); 4Service de Neurologie 2-Mazarin, AP-HP Hôpitaux Universitaires La Pitié Salpêtrière—Charles Foix, 75013 Paris, France; mehdi.touat@aphp.fr; 5Institut du Cerveau et de la Moelle Epinière, CNRS, UMR S 1127, Inserm, Sorbonne Université, 75013 Paris, France; 6Centre de Vision Numérique, OPIS, CentraleSupélec, Inria, Université Paris-Saclay, 91190 Gif-sur-Yvette, France; emilie.chouzenoux@centralesupelec.fr (E.C.); arnaud.quillent@inria.fr (A.Q.); 7Department of Radiation Oncology, Gustave Roussy Cancer Campus, 114 Rue Edouard Vaillant, 94800 Villejuif, France; ELAINE.LIMKIN@gustaveroussy.fr; 8Department of Radiology, New York Presbyterian, Columbia University Irving Medical Center, New York, NY 10032, USA; ld2752@cumc.columbia.edu; 9Medical Oncology Unit, CHU de La Réunion, Reunion University, 97410 Saint Pierre, France

**Keywords:** glioblastoma, bevacizumab, biomarker, radiomics, machine learning

## Abstract

Anti-angiogenic therapy with bevacizumab is a widely used therapeutic option for recurrent glioblastoma (GBM). Nevertheless, the therapeutic response remains highly heterogeneous among GBM patients with discordant outcomes. Recent data have shown that radiomics, an advanced recent imaging analysis method, can help to predict both prognosis and therapy in a multitude of solid tumours. The objective of this study was to identify novel biomarkers, extracted from MRI and clinical data, which could predict overall survival (OS) and progression-free survival (PFS) in GBM patients treated with bevacizumab using machine-learning algorithms. In a cohort of 194 recurrent GBM patients (age range 18–80), radiomics data from pre-treatment T2 FLAIR and gadolinium-injected MRI images along with clinical features were analysed. Binary classification models for OS at 9, 12, and 15 months were evaluated. Our classification models successfully stratified the OS. The AUCs were equal to 0.78, 0.85, and 0.76 on the test sets (0.79, 0.82, and 0.87 on the training sets) for the 9-, 12-, and 15-month endpoints, respectively. Regressions yielded a C-index of 0.64 (0.74) for OS and 0.57 (0.69) for PFS. These results suggest that radiomics could assist in the elaboration of a predictive model for treatment selection in recurrent GBM patients.

## 1. Introduction

Glioblastoma (GBM), the most common malignant primary intracranial tumour in adults, is associated with a dismal prognosis and a median survival less than 12 months [1,2]. The current standard of care for newly diagnosed GBM is based on maximal safe resection followed by adjuvant radiation therapy and chemotherapy (oral temozolomide) [3,4,5]. Unfortunately, despite this multimodal strategy, almost all patients will eventually relapse; the 5-year survival is merely equal to 10% [6,7,8,9]. GBM, known to be a highly vascularized tumour, generates excessive levels of vascular endothelial growth factor (VEGF), a crucial component of tumour angiogenesis. Bevacizumab, an antiangiogenic drug that targets VEGF ligands [10,11,12], has shown significant activity in GBM and is considered as a valid option by the Food Drug Administration (FDA) for the treatment of recurrent glioblastoma [12,13]. Bevacizumab has significantly improved progression-free survival (PFS) when added to the standard protocol. However, both phase II and III clinical trials failed to demonstrate any significant impact on overall survival (OS) [13,14,15,16,17]. Nevertheless, the survival outcomes remain very heterogeneous, and a subgroup of patients, with a potential clinical benefit from bevacizumab, has yet to be defined [18,19].

With the paucity of options in the recurrent setting, bevacizumab remains largely used in GBM patients due to the improvement in their quality of life. There is an unmet need to identify those patients who might benefit from this drug to prevent unnecessary outcomes and side effects among the entire population. Recently, advances in neuro-imaging with the complex data acquirement (functional and morphological data) from GBM patients have led to a tremendous evolution in the field. The use of artificial intelligence (AI) has largely facilitated these analyses and the evaluation of imaging to better predict outcomes [20]. With the clinical and survival data of GBM as well as the molecular and genetic alterations, generating mathematical methods, with the help of radiomics, for the proper prediction of response to therapy can become a reality in the personalized medicine era.

Radiomics, an emerging field in radio-diagnostics, consists of extracting a high number of quantitative data from radiographic medical images using specific data algorithms. The features include the tumours’ shape, texture, and voxels’ intensity, which have the potential to uncover patterns that the naked eye could not detect on standard images. Previous experiences have successfully used the radiomics’ features in the prognostic prediction of GBM patients treated with bevacizumab [21,22,23,24], but still further analysis on different datasets is needed to draw a complete conclusion on the potential use of radiomics in GBM survival analysis. Moreover, recent work has successfully used a radiomics model in the prediction of the response to bevacizumab in patients with brain necrosis after radiotherapy [25]. In this paper, we evaluated the potential of machine-learning (ML) algorithms based on radiomics in combination with clinical features for generating predictive mathematical models capable of predicting the survival outcomes of recurrent GBM patients treated with bevacizumab.

## 2. Materials and Methods

### 2.1. Patient Selection

Clinical and radiological data were retrospectively collected from magnetic resonance imaging (MRI) tests performed on 200 patients with recurrent GBM (confirmed histologically) at Gustave Roussy Cancer Campus (Villejuif, France) between 2006 and 2016 (based on the World Health Organization (WHO) classification of central nervous system tumours, Grade IV [26]). All enrolled patients must have received bevacizumab for the treatment of recurrent GBM after a first-line treatment failure, which consisted most often in surgery (72% patients) followed by post-operative chemoradiotherapy or chemoradiotherapy alone. Patients under 18 years or above 80 years of age were excluded from the analysis. Further cohort characteristics are detailed in Table 1.

The study was approved by the institutional review board per RGPD provisions. The study was declared on the Health Data Hub site and the CNIL per RGPD recommendations. Additionally, all patients were informed of their enrolment in the study.

### 2.2. MRI Protocol

MR acquisitions were all performed on 2 imaging machines (MRI) from the same manufacturer (General Electric^®^, Milwaukee, USA): Optima MR450w 1.5T and Discovery MR750w 3T. MRI data included at least: a post-contrast (gadoterate meglumine Dotarem, Guerbet, Villepinte, France), a three-dimensional T1-weighted fast spoiled gradient recalled (FSPGR) acquisition (post-contrast 3DT1), post-contrast 3DT1, and fat-suppressed FLAIR images. Only MR images were used as inputs of the radiomics classifier. To ensure image quality, neuroradiologists analysed all the available imaging sequences. Table 2 details the MRI parameters for both machines.

### 2.3. Image Analysis

#### 2.3.1. Pre-Processing

MR image pre-processing included: Z-score normalization and spatial resampling (with a target voxel size of 1 × 1 × 1 mm^3^) that was performed due to data inhomogeneity.

#### 2.3.2. Tumour Segmentation

Segmentation of two volumes of interest (2D) were performed semi-automatically using Olea Sphere^®^ (Olea Medical). The first volume was segmented on post-contrast T1 images and included the enhancing area and the necrotic regions, while the second segmentation was executed on the FLAIR images and included the FLAIR hypersignal. Within a region of interest defined by a trained radiologist (AS with 10 years of experience), the threshold-based grey level contouring and the manual correction were used for the segmentations so that the volumes of interest would be carefully drawn along both tumour enhancement and FLAIR hypersignal.

#### 2.3.3. Radiomics and Features Extraction Technique

The whole radiomic pipeline is summarized in Figure 1. Radiomic features were extracted from the segmented T2 FLAIR and gadolinium-enhanced images using the Olea Sphere^®^ (Olea Medical La Ciotat, France) radiomics package. An absolute discretization with a bin size of 37 was chosen. This was equivalent to a 32 fixed bin number based on the mean of the intensity intervals computed for all patient volumes of interest within the dataset. Several feature classes were considered: 19 first-order statistics, 17 shape-based features, 23 grey level co-occurrence matrix features (GLCM, texture), 16 grey level run length matrix features (GLRLM, texture), 15 grey level size zone matrix features (GLSZM, texture), 5 original neighbouring grey tone difference matrix, and 14 grey level dependence matrix features (GLDM, texture). To remove potential non-biological variations related to different scanning machines, the Combat normalization from the NeuroCombat Python [27,28] package was applied to the radiomic features. In addition to the radiomics, 9 baseline clinical characteristics (age, sex, delay between the diagnosis and the start of treatment with bevacizumab treatment (Delay R), surgery (yes/no), presence of symptoms (nausea/vomiting, fatigue...), neurological deficit, epilepsy, intracranial hypertension (ICH), or haematological abnormality) before bevacizumab treatment initiation were also extracted.

#### 2.3.4. Machine-Learning (ML) Algorithms

Survival analysis was performed to predict the OS and PFS; then, binary classification models were created at different endpoints. For the classification models, 7 different classifiers were trained: random forest (RFT), gradient boosting, Adaboost, logistic regression, K-neighbours, naïve Bayes, and SVM. All the algorithms were taken from the Scikit-learn Python library. Survival analysis differed from classical regression by the existence of censored data. Survival random forest [29,30] is a modified random forest algorithm that can perform such an analysis by calculating the survival function. This function gives the probability that one patient survives longer than a specific time. It can also calculate the risk score, which is a value computed on an arbitrary scale. Then, if samples are ordered according to their predicted risk score (in ascending order), one obtains the sequence in time of events, as predicted by the model. We used the RandomSurvivalForest from the Scikit-survival Python library to perform such a regression.

#### 2.3.5. Model Building

The pipeline for the model building is summarized in Figure 2. Within the cohort, 6 patients were excluded because of missing clinical data. Hence, the data from 194 patients were used for the regressions. Since 44 patients had a censored date of death, 8 of them were censored after 15 months. Therefore, 158 patients were included for the 15-month survival classification. The same logic was applied for the other classification models (Table 3). For each model, patients were divided into a training (80%) and test cohort (20%) (Table 3). The proportions of each class over the entire cohort were kept in the training and test sets. Only the training set was used for the development of the statistical models. First, cross-validation (CV = 5) was applied to the training set to select the best hyperparameters for each ML classifier and the selection of the best one to be tested on the test set. Then, the chosen classifiers were trained on the overall training dataset and validated on the test set.

Before the training, feature processing was applied using only the training set. First, the variable was standardized using the StandardScaler from the Scikit-learn Python library. Then, because of the high number of features compared to the number of patients and to overcome the well-known “curse of dimensionality” in ML, we performed a feature selection to reduce dimensionality and decrease the risk of overfitting: only the features with a concordance index (C-index) with the OS strictly superior to a certain threshold value were kept. Then, if two of the remaining variables had a covariance superior to 0.5, the one with the lowest C-index was eliminated. Afterward, the standardization model and the feature selection were applied to the test set. The C-index threshold was chosen according to the best result within the cross-validation. The same feature processing was implemented on the training set and applied to the validation set for every fold of the CV. Hence, several lists of features were obtained, depending on the train-fold utilized. Performing the feature selection on the overall dataset would introduce bias in the results, as we would use the samples within the test sets to select them.

#### 2.3.6. Resampling

Due to the occurrence of imbalanced classes in the classification models (Table 4), a resampling technique was used to avoid overfitting of the majority class. This family of techniques has already been widely studied to deal with imbalanced datasets [31,32]. It consists of generating new data in the training set to have an equivalent amount of data within each class. The RandomOverSampler resampling method from the imblearn Python package was performed. This function creates new samples by reproducing already existing data in the under-represented class. We set the shrinkage parameter to 1.3 so that the new sample can differ only a little from the original one. The SMOTE resampling method has also been tried but did not lead to better results. After resampling, the same number of samples were obtained in each class of the training sets.

### 2.4. Statistical Analysis

OS was calculated as the number of days between the initiation of bevacizumab treatment and death. PFS was calculated as the number of days between the start of the treatment and progression or death. PFS was evaluated on all images by a neuroradiologist with 10 years of experience using FLAIR and 3D T1 sequences with gadolinium injection according to the RANO recommendations in association with a clinical evaluation (by a neuro-oncologist). We used the concordance index (C-index) as an evaluation metric for the regression. The C-index was defined as the ratio between the number of concordant pairs of predicted values divided by the total number of pairs of patients (this value does not take into account the pairs including censored data or an event taking place after the censoring). The area under the curve (AUC) of the receiving operating characteristic (ROC) curve was used to evaluate the classification performance. The classifiers’ selection was based on the best mean metric score performed on the 5-fold of the cross-validation. Kaplan–Meier curves and log-rank statistic test were used to assert the building of two significantly different populations by the binary classification in terms of survival.

## 3. Results

### 3.1. OS Outcomes

Several ML algorithms were tested to determine the method best suited to the cross-validation. For the 9-month classification, a logistic regression insured the best performance with an AUC on the test sets equal to 0.78 (0.79 on the train). An SVM with a Gaussian kernel had an AUC of 0.85 (0.82) in the 12-month classification, while an RFT had an AUC of 0.76 (0.87) in the 15-month model. The SFT obtained a C-index of 0.64 on the test set (0.74 on the train). It performed a mean score of 0.63 ± 0.06 (±1 standard deviation) on the cross-validation. The logistic regression accomplished a mean AUC of 0.62 ± 0.12 for the 9-month models, while the SVM had a score of 0.71 ± 0.20 for 12 months and the RFT a score of 0.60 ± 0.10 on the cross-validations. Precision and recall for each class of each classification model are provided in Table 5. Cross validation results for every classifier and classification model are provided in Appendix A.

The Kaplan–Meier curve showing the survival probabilities (Figure 3) of the two classes predicted by the classifier on the test depicts two well-separated curves (log-rank *p* < 0.005) on the 9 months model. The class “beyond 9 months” obtained a survival probability at 6 months of 0.85, while the class “before 9 months” obtained a probability of 0.45. These probabilities went down to 0.71 versus 0.25 at 10 months and 0.43 versus 0.15 at 20 months. Without the resampling method, the two other models tended to classify all patients in the majority classes. Hence, well-balanced classes or a more adapted resampling algorithm for this study could improve these outcomes. Nevertheless, these two models also succeeded in the identification of two different populations with a significant difference in the survival probability distributions (log-rank *p* = 0.01 for both models). For instance, the 12-month classification identified all the patients who did not survive after the endpoint except two (15 patients out of 17). Moreover, the class “after 12 months” had more than a 0.5 probability of surviving beyond 1 year. Such a model could offer a great perspective into the prediction of prognostic value.

### 3.2. PFS Outcomes

The regression model obtained a C-index value of 0.57 on the test set and 0.69 on the training set. The two classification models for progression at 6 and 9 months failed to stratify the cohorts in two significantly different populations according to the log-rank test (*p*-value > 0.05). These models obtained an AUC, respectively, equal to 0.56 (0.71) and 0.69 (0.82). Recall, precision (Appendix A), and Kaplan–Meier curves (Appendix A) are available in the Appendix A.

### 3.3. Clinical and Radiomic Signature

On the four different models for OS (regression and classifications), on average, nine features were selected on the training sets each time (eight for 9 and 12 months, nine for 15 months, and twelve for the regression). Four of these features were constantly present in all selections: one histogram intensity feature from the FLAIR images (original first-order 10th percentile), one shape feature from the gadolinium images (original shape sphericity), and two clinical variables that were the presence of constitutional symptoms or the presence of a neurological deficit. The other selected features included delay R, gender, the original shape-surface-area-to-volume ratio (shape), and the original neighbouring grey tone difference in matrix strength (texture) from FLAIR images, which appeared in every model except one. Texture maps of original neighbouring grey tone difference in matrix strength radiomic of one patient with good prognosis and one with bad prognosis are provided in Figure 4.

According to the C-index of these variables, none of them seemed essential for the prognostic value (Table 6). All the C-index ranged between 0.55 and 0.61. Only the shape sphericity had a value more than 0.60 for the 12-month classification. For the PFS regression, 11 features were selected including the presence of constitutional symptoms, the presence of a neurological deficit, FLAIR’s area-to-volume ratio and difference matrix strength, and Gadolinium’s sphericity. The eight features tended to show higher values for patients with good prognosis. The higher the features, the higher the chances for a patient to have a longer survival.

## 4. Discussion

In this paper, we evaluated the ability of several ML algorithms, based on radiomics from multimodal imaging in combination with clinical characteristics, to predict survival in recurrent GBM patients treated with bevacizumab. Nine different clinical variables, 109 radiomics from T2 FLAIR, and 109 radiomics from gadolinium-enhanced MRI images from a cohort of 194 patients were included in the analysis. Our results suggest that these models can provide valuable insights into the OS of recurrent GBM patients. The novel classification models successfully stratified patients into two significantly different populations, thus suggesting a possible role in the selection of patients who might benefit from bevacizumab in the recurrent setting.

The survival analysis was performed at three different endpoints at 9, 12, and 15 months after the beginning of the bevacizumab treatment for the OS analysis and at 6 and 9 months for the PFS. These time limits were pre-specified after being considered to be relevant according to the neuro-oncologists for their utility in clinical practice. Grossman et al. [21] already used these 6- and 9-month endpoints for PFS and the 12-month endpoint for OS in their analysis. Additionally, in comparison to other survival analyses that stratified the patients into low- and high-risk cohorts using the median survival [22] or the median predicted risk function [23], the selected endpoints in this paper should provide fewer cohort-specific and more generalizable models. Kickingereder et al. [23], who used radiomics from pre- and post-contrast T1-weighted images (referred to as T1 and cT1) and FLAIR sequences at baseline, have successfully stratified the patients into low and high risk (groups assessed by predicted risk function and cox regression analysis) for PFS. Nevertheless, their model was more accurate in terms of OS prediction rather than PFS. These differences between PFS and OS could be explained by the existence of a pseudo-response, a pseudo-progression [33,34,35], or the fact that the PFS endpoint could not be well defined.

The redundancy in the selection of features within every model could lead to the identification of a radiomic and clinical signature for the prediction of survival. Previous studies have already shown the ability of radiomics, extracted from T1-weighted and FLAIR imaging, of pre-therapy prediction of prognosis [21,23], but only Grossman et al. also identified a specific feature signature that was composed of 10 texture radiomics from T1 images and 10 texture and shape features from the FLAIR images. Moreover, it appeared in their model that T1-weighted images (and, more precisely, texture information) provided more information on survival prediction than FLAIR images where only shape features seemed to be significantly correlated with PFS and OS. The difference with the signature in this paper could be explained by their feature selection method. Indeed, they used a principal component analysis to select their best features. This method is based on the hypothesis that the best features are the ones that explain the variations within the data. The C-index method can select the features that allow the correct order of patients according to their date of death. This last method seemed more appropriate for survival analysis. In addition, they selected independently on each MRI modality (FLAIR and T1) the top 10 features, while we have carried out the selection on both modalities at once. The selection of the optimal features is a crucial step in the adaptation of radiomics that have drastically impacted our models with the support of the C-index threshold. For instance, Chang et al. [22] who used the top predictive 128 features among 2293 pre-therapy radiomics features, failed to stratify their cohorts on the test sets using a random forest. Grossman et al., who predicted PFS and OS on similar endpoints as us, used a similar number of features in their random forest. Their model stratified two populations for PFS at 3, 6, and 9 months and OS at 12 months.

The clinical variables such as the delay between the diagnosis and the first relapse, the presence of symptoms, or the presence of neurological deficits at the initiation of the therapy seem to be quite correlated with survival outcomes, as suggested by their C-index with OS. To our knowledge, no other studies ever used these features. Moreover, the correlation threshold in this analysis suggests that these clinical variables offer complementary data to the radiomics, which may lead to the elaboration of other advanced models using different sources of data. Various studies have shown that the apparent diffusion coefficient (ADC) from diffusion MR could represent a good predictor of survival in GBM patients treated with bevacizumab [36,37,38,39]. In addition, perfusion-derived biomarkers were explored [40,41,42], but it should be mentioned that these methods depend largely on the type of the MRI machine, the sequence processing software, and the calculation algorithms and, therefore, are more difficult to apply to different centres in routine clinical practice. Non-MRI-based biomarkers from transcriptomic [43,44,45] or the neutrophil-to-lymphocyte ratio [46] have also been studied for the prognostic of patients treated with bevacizumab. Although, they have been able to stratify two significantly different populations in term of survival outcomes, to our knowledge, none of them used as many patients as in our study and succeeded to accurately predict the OS or PFS. Using different sources of data could improve the accuracy and robustness of the predictive models in comparison to radiomics-only models.

Nevertheless, onemajor limitation of radiomics are robustness and reproducibility. During the extraction from the manual or semi-automatic tumour segmentation, it is impossible to reproduce the same contours twice. In particular, the robustness of the shape feature sphericity must be verified over different segmentations to be able to assert the generalization of the model. On the other hand, our most significant features according to their C-index with OS on training sets are logical and simple parameters that can be easily adapted in comparison to the other signatures described in the literature on texture and automotive prognosis. Another important finding was the signal intensity of the tumour in the FLAIR images that reflected the cellularity contour and volume, known to have a poor prognosis in large lesions, while the delay between the diagnosis and the start of treatment with bevacizumab treatment (Delay R), the presence of symptoms, or the presence of neurological deficit largely reflected the aggressiveness of the tumour.

Although previous data have demonstrated that OS is not significantly improved with bevacizumab, further studies are needed to confirm the reproducibility of our model with the possibility of stratifying a cohort of GBM not previously treated with bevacizumab into two distinct populations. That way, a sub-cohort of patients that would benefit from bevacizumab can be accurately identified. Indeed, a connection between patients with a good prognosis and the response to bevacizumab treatment cannot be confirmed, since they could have a good prognosis regardless of the therapy. In this paper, the cross-validation failed to reach high accuracy, and huge differences were observed, which could be related to the relatively small sample size. Nevertheless, these results could lead the way to further analysis that could aid in the development of a prediction model based on radiomics in daily clinical practice. Predictive biomarkers have become a major component of cancer management to guide the therapeutic strategy where treatment selection for each patient has become personalized [47]. After further re-evaluation of our models, with the adaptation of ML models based on radiomics and clinical data, a decision tool could be developed to select those GBM patients who are most likely to benefit from bevacizumab. On the other hand, this tool may also identify those patients who are less likely to respond to bevacizumab and might require an alternative or a more aggressive approach. These advances in the radiomics field in GBM and the implementation of predictive models in daily practice will eventually encourage researchers to invest their efforts in the adaptation of similar models in other malignant tumours.

In conclusion, radiomics in combination with clinical data have successfully predicted the survival of recurrent GBM patients treated with bevacizumab. Three different binary models for survival prediction at 9, 12, and 15 months with high performances were built, which could lead to the creation of a convenient tool for decision making and the orientation to a more patient-specific treatment in the era of personalized medicine. Larger trials are needed for better identification and adaptation of these models in GBM patients.

## Figures and Tables

**Figure 1 diagnostics-11-01263-f001:**
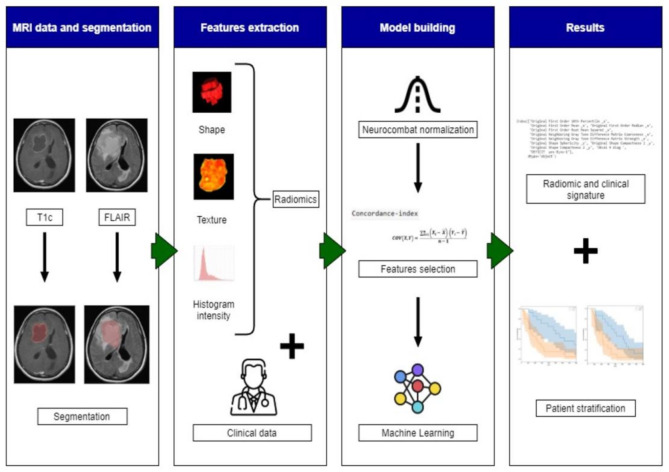
The workflow of radiomics and analysis used in this study. The overall procedure of identifying a MRI radiomics signature model and a practical ML model for stratifying the GBM patient’s prognosis based on OS.

**Figure 2 diagnostics-11-01263-f002:**
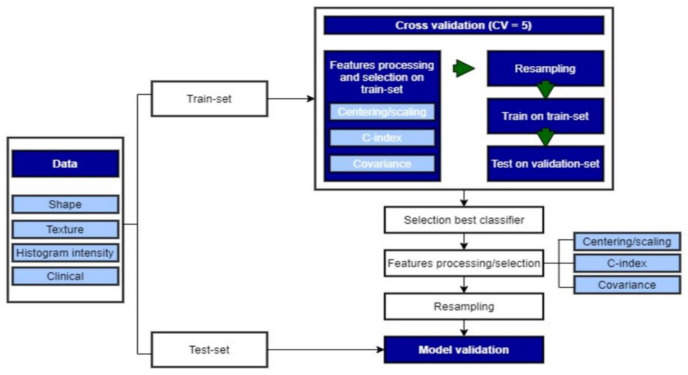
Model building using ML.

**Figure 3 diagnostics-11-01263-f003:**
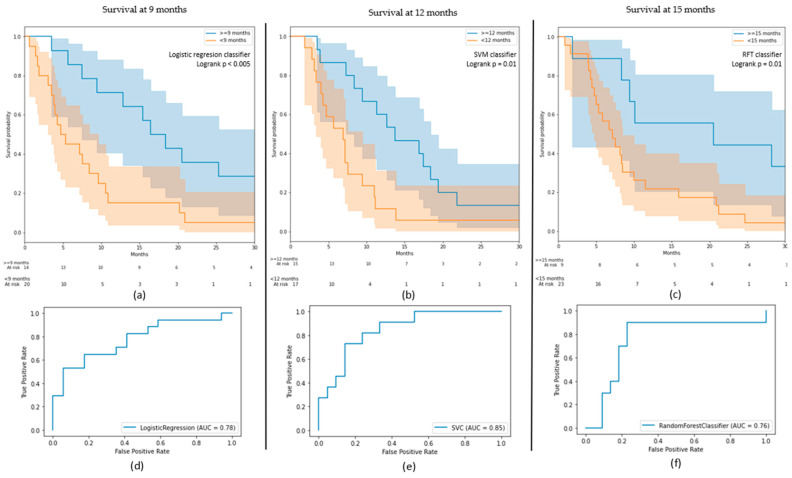
(**a**–**c**) Kaplan–Meier curves for the 9-, 12-, and 15-month models; (**d**–**f**) receiver operating characteristic (ROC) curves of the results on the test sets for the 9-, 12-, and 15-month models.

**Figure 4 diagnostics-11-01263-f004:**
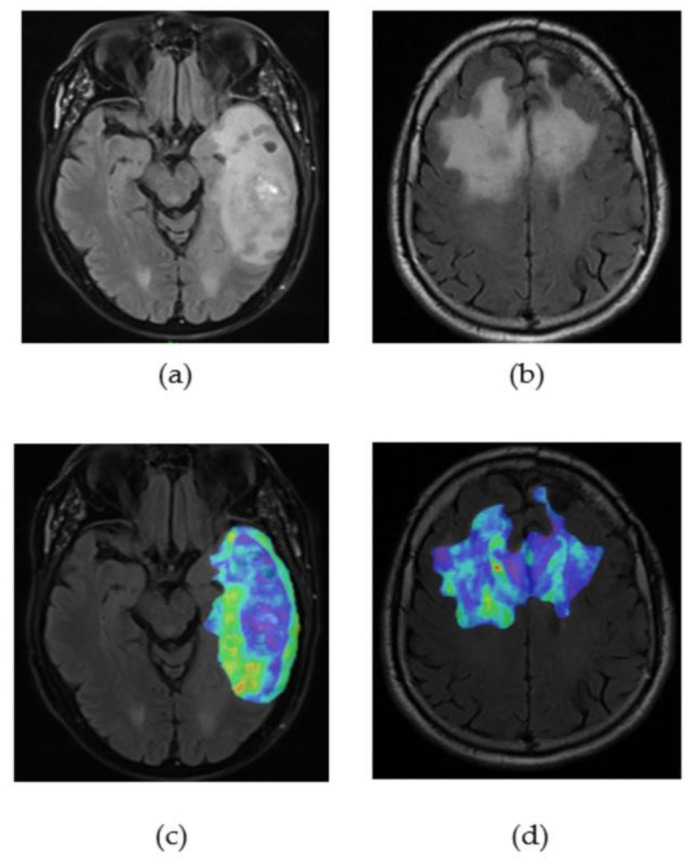
(**a**) FLAIR image of glioblastoma’s patient with a bad prognosis (OS = 3 months); (**b**) FLAIR image of glioblastoma’s patient with a good prognosis (OS = 30 months); (**c**) original neighbouring grey tone difference in matrix strength radiomic map of the segmented tumour on the patient with bad prognosis; (**d**) original neighbouring grey tone difference in matrix strength radiomic map of the segmented tumour on the patient with good prognosis.

**Table 1 diagnostics-11-01263-t001:** Patient characteristic.

**Characteristics**	**Value**
Age----------year	
Mean	56.6
Median	58
Min–max	18–80
Sex----------n (%)	
Female	74 (37%)
Male	126 (63%)
OS----------days	
Mean	357
Median	244
Min–max	19–2686
PFS----------days	
Mean	294
Median	204
Min–max	16–2686
Tumour location----------n (%)	
Left	93 (45%)
Right	93 (47%)
Multifocal	16 (8%)
**Clinical Variable ^1^**	**Value**
Delay R ^2^----------days	
Mean	415
Median	327
Min–max	23–2202
Surgery---------- n (%)	
Yes	143 (72%)
No	57 (28%)
Symptoms---------- n (%)	
Yes	164 (78%)
No	44 (22%)
Neurological deficit---------- n (%)	
Yes	68 (34%)
No	132 (66%)
Epilepsy---------- n (%)	
Yes	26 (13%)
No	174 (87%)
Intracranial hypertension ---------- n (%)	
Yes	41 (21%)
No	159 (79%)
Haematological toxicity---------- n (%)	
Yes	48 (24%)
No	152 (76%)

^1^ Patient data retrieved from clinical examination at the start of the bevacizumab treatment. ^2^ Delay between GBM diagnosis and start of bevacizumab treatment.

**Table 2 diagnostics-11-01263-t002:** MRI parameters.

Machine	Weighting	Sequence	TR	TE	Slice Thickness
Optima MR450w 1.5 T Installed in 2016, 70 cm tunnel, 32 channels, 40 cm z-axisFOV, gradient 40 mT/m SR 200 T/m/s.	T1 pre-contrast	3D rapid gradient echo	9 ms	4.2 ms	1 mm
	T2-FLAIR	Turbo spin echo	7002 ms	138 ms	1.4 mm
	DWI	EPI, two b-values (0 and 1000 mm/s)	3349 ms	79 ms	4 mm
	T1 post-contrast	3D rapid gradient echo	6.1 ms	1.2 ms	1 mm
Discovery MR 750w 3 T Installed in 2012, 70 cm tunnel, 32 channels, 50 cm z-axisFOV, gradient 44 mT/m SR 200 T/m/s.	T1 pre-contrast	3D rapid gradient echo	9 ms	2.1 ms	1 mm
	T2-FLAIR	Turbo spin echo	7002 ms	118 ms	1 mm
	DWI	EPI, two b-values (0 and 1000 mm/s)	3349 ms	62.6 ms	3 mm
	T1 post-contrast	3D rapid gradient echo	6.1 ms	2.1 ms	1 mm

**Table 3 diagnostics-11-01263-t003:** Training and test set patient repartition.

Model	Train–Test	Total
Survival regression	155 39	194
9 months survival	132	166
	34	
12 months survival	126	158
	32	
15 months survival	126	158
	32	
6 months progression	147 37	184
12 months progression	147 37	184

**Table 4 diagnostics-11-01263-t004:** Classes’ distribution before resampling.

Model	Classes	Number (%)
9 months survival	≥9	81 (49%)
	<9	85 (51%)
12 months survival	≥12	56 (35%)
	<12	102 (56%)
15 months survival	≥15	47 (30%)
	<15	111 (70%)

**Table 5 diagnostics-11-01263-t005:** Results metrics of the classification models on the test sets.

Model	Best Classifier	AUC on Test (on Train)	Classes	Precision	Recall
9 months survival	Logistic regression	0.78 (0.79)	≥9	0.70	0.82
			<9	0.79	0.65
12 months survival	SVM	0.85 (0.82)	≥12	0.88	0.60
			<12	0.71	0.82
15 months survival	RFT	0.76 (0.87)	≥15	0.78	0.82
			<15	0.56	0.50

**Table 6 diagnostics-11-01263-t006:** Essential features based on their C-index with the OS on the training set sets.

Feature	Source	Radiomic Type	9 Months (C-Index)	12 Months (C-Index)	15 Months (C-Index)	Regression (C-Index)
1st-order 10th percentile	FLAIR	Histogram intensity	0.59	0.56	0.56	0.59
Sphericity	Gadolinium	Shape	0.59	0.60	0.57	0.59
Deficit	Clinical	-	0.56	0.56	0.58	0.59
Symptoms	Clinical	-	0.56	0.58	0.57	0.57
Delay R	Clinical	-	-	0.55	0.59	0.55
Area-to-volume ratio	FLAIR	Shape	0.57	0.57	0.55	-
Difference matrix strength	FLAIR	Texture	0.57	0.57	-	0.59
Sex	Clinical	-	0.57	-	0.56	0.59

## Data Availability

The data presented in this study are available on request from the corresponding author. The data are not publicly available due to privacy and ethical concerns.

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
