# Peer review of "Machine-Learning-Based Radiomics MRI Model for Survival Prediction of Recurrent Glioblastomas Treated with Bevacizumab"

_diagnostics, 2021, doi:10.3390/diagnostics11071263_

Round 1

Reviewer 1 Report

The paper turns out well written with adequate English. The use of radiomics in the study of gliomas appears to be a very interesting field of research. I find very interesting to include in the radiomic analysis also the FLAIR signals in addition to the T1-weighted sequence with mdc. However, I am highly dubious about the indistinct use of images evaluated with 1.5T and 3T machinery, how can one make sure that there is no bias provided by this difference in imaging? The results are not explained very clearly and Table 1 appears to be poorly composed. 

I recommend minor revision.

Reviewer 2 Report

Machine learning based radiomics MRI model for survival prediction of glioblastomas treated with bevacizumab

In this study the authors extracted radiomic features from CET1 and FLAIR images of recurrent GBM treated with bevacizumab, and employed them to successfully build a machine learning model capable of predicting the overall survival and progression free survival.

The clinical question behind the study is clear and valid: although bevacizumab (an anti-angiogenetic drug) has been extensively employed in recurrent glioblastomas, the results are still somewhat unconvincing specially in terms of overall survival. Therefore, there is an unmet need for biomarkers that may help selecting a subset of patients that may actually benefit from this treatment.

The radiomics method applied by the authors looks solid. All the steps are part of the state-of-the-art radiomics workflow: normalization and resampling, CET1-FLAIR tumor segmentation, feature extraction, feature normalization, training several classifiers, cross-validation. In addition, applying a resampling step preventing imbalanced classes to affect the training sounds like a good idea.

The main finding of the article is that radiomics metrics can be used to select a subset of patients that may benefit more from bevacizumab.

Overall, the work looks solid and valid. However, I believe that some modifications may improve the manuscript, and should be at least considered prior to publication. I am writing down all the modifications that I suggest, but, while some of them are just suggestions to make the paper more clear and effective (PART B), other ones are crucial to be clarified prior to publication (PART A: Major concerns). Finally, PART C features minor comments that can be fixed quickly, if appropriate.

PART A – Major concerns

1) Patient selection Line 87: how long did the patients receive bevacizumab after recurrence? This is a crucial question to address, since patients were followed for >15 months (many of them), so it is important to understand whether some of them discontinued the therapy and/or underwent other therapies afterwards. Did all of them continue bevacizumab for the whole follow-up time?

2) Tumor segmentation Line 113-118: if my understanding is correct, two separate tumor masks were generated: 1) T1 enhancement mask; 2) FLAIR signal alteration. In many radiomics studies, these two masks are generated, and usually the first one includes enhancing areas without the necrosis areas, and the second one includes all of the surrounding non-enhancing FLAIR alterations (see for instance Figure 3 in Gian Marco Conte et al, Radiology 2021 https://pubs.rsna.org/doi/full/10.1148/radiol.2021203786). Was this the case? This should be specified, since the results in the radiomics analysis can vary dramatically depending on the segmented tissue. Also, in case the two masks were generated as in Conte et al, Figure 1 of this present paper would be very misleading, since the CET1 mask includes the necrosis area and the FLAIR mask is not displayed (the mask overlayed with the FLAIR images is the same as the CET1).

3) Table 5. It is very useful to have this lookup table which features were more determinant for the prognosis stratification. However, the authors are not stating whether these features showed higher/lower values in patients with better/worse prognosis: was the 10th percentile FLAIR higher in patients with bad or good prognosis? Tumors with more ‘spherical’ shape had better or worse prognosis? Tumors with a higher area-to-volume ratio had better or worse prognosis? To answer this, it would be sufficient to add one column in the table that states whether the single metric was higher in subjects with better or worse prognosis.

4) Limitations: the study did not include any patients that was not treated with bevacizumab. Is it correct to state that perhaps the algorithm can distinguish bad prognosis GBM from good prognosis GBM regardless of the treatment? That is, is it correct to state that there is a chance that some tumors that showed good prognosis would have good prognosis regardless of the type of therapy? Can the authors comment on this and, only if appropriate, insert a comment in the discussion/limitations section?

PART B – Additional suggested modifications

1) OS outcomes, section 3.1: this section includes the main findings of the study, with many numbers and combinations (3 different classes, each one with a classifier, and each has different values of AUC, C-index, log rank p value, precision …). I would consider producing a table that includes all these values with the corresponding class and classifier, so that the message can be clearer also for readers that are not completely experienced with radiomics analyse. Also, would it be possible to obtain a pooled OS value in terms of months? That would make the results easier to read by an oncologist, and more comparable to other prior studies (see for instance Kickingereder et al “Large scale….”, and see also the review “Bevacizumab and glioblastoma” by Kim et al, where OS in terms of months are reported from several studies).

2) Figure 3. This figure represents the most important finding of the paper, I would consider expanding it and giving it more importance. If my understanding is correct, panel A “survival at 9 months” corresponds to a logistic regression, panel b “12 months” to SVM, panel c “15 months” to RFT – I would specify these corresponding classifiers in the figure and report the p-value. Also, it would be interesting to see the corresponding ROC curves (e.g. panel D showing the ROC curve for the logistic regression representing survival at 9 months, placed underneath panel A; then panel E with the ROC curve for SVM underneath panel B; and so on).

3) Figure 4. This figure does not seem to convey any remarkable pieces of information. While it is nice to actually visualize a map of radiomic features on a MRI scan, a single case on its own does not show anything interesting in my opinion. On the contrary, it would be nice to visualize a difference in one significant MRI features between a representative patient with bad prognosis and a representative patient with good prognosis. This could be either on a MRI scan (like shown in the single subject of figure 4 already) or with a histogram (perhaps showing that the 10th percentile FLAIR is very different between good and bad prognosis, for instance)

PART C – Minor considerations

  • Title: only recurrent GBM are included, consider modifying the title into “…survival prediction of recurrent glioblastomas treated with bevacizumab”.
  • Abstract Line 33: patients >80 years were excluded, so I would state this in the abstract Line 33 [e.g. “recurrent GBM patients (age range 18-80)”].
  • Abstract Line 32: the patients included in the analysis were 194, consider stating 194 in the abstract instead of 200.
  • Abstract Line 39: “model usable in the clinical practice”, consider soften this statement since the model must be validated multiple times before a clinical application.
  • Introduction Line 74: “none have evaluated its impact on the prediction of PFS and OS”. If my understanding is correct, this sentence is not correct and should be changed. For instance, in the paper by Kickingereder 2016 (Large scale….) the authors demonstrate that their radiomics model can distinguish between a high-risk subset of patients (PFS/OS 3.8/6.5 months) and a low-risk one (5.9/11.8), so that it is clear that it has already been demonstrated that radiomics can predict PFS and OS specifically in the setting of bevacizumab-treated GBM.
  • Patient selection Line 94: how could the patients be informed of their enrollment in this particular study given that the study was retrospective (Line 83)? Maybe they were informed that their images would have been used for research purposes? If so, it is fine, but the sentence should be rephrased.
  • Table 1: please specify the abbreviations in the caption of the table.
  • Table 2: please consider reporting the voxel size (e.g. 1x1x1 mm) rather than the slice thickness only.
  • Table 2: why are T1 pre-contrast and DWI in this table? These sequences were not employed in the analysis, correct?
  • Pre-processing Line 110: spatial resampling, what is the target voxel-size?
  • ML algorithms Line 147: “the information that they have been alive…” this sentence is not very clear in my opinion.
  • Statistical analysis Line 200-201: how was progression defined? Did a neuroradiologist score all the follow-up scans in order to define progression? Did they use the RANO criteria?
  • Figure 4 caption: “93 days”, not “93 jours”
  • Discussion Line 279: again consider stating 194 patients instead of 200, as in the abstract
  • Discussion Line 288: “6 and 9 month endpoint” the word “month” is missing – typo
  • Discussion: “these results constitute an essential backbone for the development…” consider soften this statement.

Strengths of the study

The radiomics method applied looks solid, and the results further support the already widespread idea of radiomics metrics representing potential in vivo markers that may guide clinical decision-making in the future.

Limitations of the study

The results are not entirely innovative, since previous studies already showed that radiomics can successfully predict prognosis of bevacizumab-treated GBM. Nonetheless, this article serves as a further confirmation in a different cohort.
